# Radial Artery Deviation and Reimplantation (RADAR) to Accelerate the Maturation of Radial-Cephalic Fistulas for Hemodialysis in Patients with End-Stage Renal Disease

**DOI:** 10.3390/jcm12206481

**Published:** 2023-10-12

**Authors:** Tomasz Gołębiowski, Mariusz Kusztal, Andrzej Konieczny, Maciej Gołębiowski, Krzysztof Letachowicz, Dariusz Janczak, Magdalena Krajewska

**Affiliations:** 1Department of Nephrology and Transplantation Medicine, Wroclaw Medical University, Borowska 213, 50-556 Wroclaw, Poland; mariusz.kusztal@umw.edu.pl (M.K.); andrzej.konieczny@umw.edu.pl (A.K.); maciej.golebiowski@student.umw.edu.pl (M.G.); krzysztof.letachowicz@umw.edu.pl (K.L.); magdalena.krajewska@umw.edu.pl (M.K.); 2Department of Vascular, General and Transplantation Surgery, Wroclaw Medical University, Borowska 213, 50-556 Wroclaw, Poland; dariusz.janczak@umw.edu.pl

**Keywords:** RADAR, arteriovenous fistula, fistula maturation

## Abstract

Background: The most common form of vascular access for hemodialysis is a native arteriovenous fistula, which connects the site of the artery to the end of the vein. The maturation process of the fistula plays a crucial role in the establishment of a functional vascular access. Radial artery stenosis is among the potential causes of maturation failure. In these cases, improving the fistula’s blood flow may be difficult, as traditional surgical reanastomosis and endovascular intervention frequently fail. Radial artery deviation and reimplantation (RADAR) is a novel and effective technique for creating primary fistulas with a high patency rate. The main disadvantage of this procedure is the ligation of the radial artery and the subsequent known consequences. Methods: To accelerate maturation, we used RADAR as a secondary approach in three patients with radial artery stenosis and maturation failure. Results: In all patients after surgery, we observed a significant increase in fistula blood flow. Two patients used fistulas for hemodialysis after surgery. We describe the image diagnosis, procedure, and benefits of this method. Conclusions: The RADAR technique may be successfully used as a secondary access in patients with maturation failure due to RA stenosis to accelerate fistula maturation.

## 1. Introduction

The maturation of an arteriovenous fistula (AVF) is a complex process involving changes in both the artery and the vein, characterized by their enlargement and a subsequent increase in blood flow. In 2019, over 80% of patients suffering from end-stage renal disease (ESRD) started hemodialysis (HD) with a catheter because of the lack of an AVF or the inability to cannulate one. Depending on age, 58.7% to 64.8% of prevalent HD patients utilized an AVF, and 15.0% to 20.2% utilized a synthetic arteriovenous dialysis graft [1]. Typically, a prolonged maturation time is associated with factors preceding AVF formation, such as vein destruction and radial artery (RA) atherosclerosis. In addition, the type of anastomosis may increase the risk of intima–media hyperplasia and juxta-anastomotic stenosis [2,3]. Finally, lower vessel sizes may result in technical errors during suturing while creating an AVF using the classic method [4]. In patients with small vessels, there is a risk of narrowing the lumen of the radial artery in the proximal portion of the anastomosis (see red point X in Figure 1), which either reduces the blood supply to the vein/fistula or causes the AVF to be supplied only by the ulnar artery.

In the case of AVF maturation failure, surgical or endovascular procedures are used. Studies have demonstrated that surgical techniques are superior to endovascular ones [5]. Surgical reanastomosis proximal to a previously created AVF and accessory vein ligation are two surgical options. Since the vessels are dilated, because of the primary anastomosis, the reanastomosis is simplified, and the side of the enlarged radial artery is connected to the end of the vein, resulting in a significantly higher blood flow than in the first AVF. The need to ligate the vein distally and the irreversible closure of the primary anastomosis are disadvantages of this technique. Naturally, side-to-side anastomosis may be performed in certain situations, saving the distal segment of the vein. In patients with coexisting proximal vein stenosis, this will be beneficial, as this distal vein portion with retrograde flow may be used for cannulation. In these situations, a Doppler ultrasonography examination with flow mapping should be carried out to evaluate the cannulation strategy. One of the disadvantages of side-to-side anastomosis is the difficulty of this technique due to the long distance between the artery and the vein and the possibility of venous hypertension with extremity edema.

It has been reported in the literature that favorable outcomes of the radial artery deviation and reimplantation (RADAR) technique have been observed, with less juxta-anastomotic stenosis and increased maturation and patency compared to standard radial-cephalic fistulas [6]. However, this approach has a significant disadvantage when performed as a primary AVF, because it requires the ligation of the radial artery, which has been criticized by some authors [7]. Nonetheless, we utilized this technique as a secondary fistula in patients whose primary fistula maturation process failed due to RA stenosis. We describe such a solution and explain why RADAR is an appropriate technique for reducing fistula maturation time in similar cases. This method was applied to three patients, and the diagnosis and procedure were described in detail.

## 2. Materials and Methods

All patients underwent preoperative clinical examination and Doppler ultrasound venous and arterial mapping using Samsung HS50 ultrasound scanner (Samsung Medison CO., LTD, Seoul, Republic of Korea). Prior to any procedure that could affect the patency of the radial artery, we used Doppler ultrasound and a modified Allen test to evaluate the patency of the ulnar artery and palmar arches. In all patients, we additionally carried out the Doppler Allen’s test and found the flow in the superficial palmar branch, dorsal digital thumb artery, and ulnar artery with radial artery compression, indicating a low risk of hand ischemia after RA closure.

The same operator (T.G.) performed all the surgeries. Intraoperatively, anastomotic Option I or II was selected based on vascular anatomy (Figure 1). Two factors played a role in the selection process: first, the length of the dissected radial artery (RA), and second, the distance between the RA and the cephalic vein (CV). The primary goal was to achieve a smooth RA arch with no abrupt angles.

### 2.1. Description of Case 1

A 38-year-old female with chronic glomerulonephritis (focal segmental glomerulosclerosis) undergoing hemodialysis for 16 months using a permanent dialysis catheter was admitted to our department due to a catheter-related infection. The patient was given standard care, consisting of an antibiotic based on the blood culture and an antibiotic lock to fill the catheter lumens. The catheter was maintained in place as a result of the patient’s prompt clinical recovery and the absence of signs of a tunnel infection. Eight months prior to admission, she had a distal radial-cephalic fistula (RCAVF), i.e., the anastomosis between the site of the RA and the end of the CV (Figure 2A). An ultrasound examination revealed a brachial artery flow of 264 mL/min and vein diameters of 5.8 mm and 5 mm in the mid-forearm and elbow fossa, respectively. The dominant vessel supplying the fistula was the ulnar artery (UA), which was 4 mm in diameter and supplied the fistula via the palmar arches (Figure 3A). The blood flow rate through the UA was 234 mL/min. Although the RA was patent, it only had a 1.3 mm cross-sectional diameter (Figure 3B). Prior to admission, many failed attempts were undertaken to cannulate the AVF. However, the blood supply was insufficient to allow effective dialysis. Considering that the primary cause of delayed maturation was the stenosis of the radial artery in the proximal portion of the anastomosis (red point X in Figure 1), it was decided that the blood supply to the CV had to be enhanced using the RADAR technique.

### 2.2. Procedure Description in Case 1

Local anesthesia consisting of 1% lignocaine solution was used during the procedure. An oblique incision was made in the distal part of the forearm, approximately 2 to 3 cm proximal to the primary fistula (Figure 2B). The appropriate vessel fragments were dissected. Over the ligature, the radial artery was severed, and the blood supply was verified by momentarily releasing the clamp. The perceived blood flow was satisfactory. The cephalic vein was then clamped both distally and proximally, and, finally, it was subsequently incised longitudinally. The vein and artery were flushed with heparinized saline, and approximately 5000 IU of unfractionated heparin was administered. Both vessels were sutured with a continuous suture, i.e., the end of the artery was anastomosed with the side of the vein using a non-absorbable polypropylene vascular suture of 6-0 (Figure 2B). During the procedure, the operator used a magnifying glass with 2.5 times magnification. Option I for anastomosis (Figure 1) was used in this case due to the long RA and the wide segment between the RA and CV. The vein filled with blood when the clamps were removed, and a typical murmur was noticed (Figure 2B). The wound was closed with skin sutures. Ultrasound examination confirmed that the fistula was patent the day following the procedure. UA and RA diameter was 4 mm and 2 mm, respectively (Figure 3C,D). The blood flow rate through the UA was 193 mL/min (Figure 3C), while the brachial blood flow rate was 482 mL/min (Figure 3E). The patient was discharged with a fistula that was functional. Neither anticoagulants nor antiplatelet medications were administered to her during the postoperative period. At the 3-week postoperative follow-up appointment, the fistula remained patent, and the first successful cannulation was performed 4 weeks after surgery (Figure 2C).

### 2.3. Description of Case 2

A 75-year-old woman with type 2 diabetes presented to our clinic due to the progression of chronic kidney disease to ESRD. Laboratory tests revealed that serum creatinine (sCr) was 4 mg/dL, urea 156 mg/dL, and potassium 4.5 mmol/L. The estimated glomerular filtration rate (eGFR) was 16 mL/min/1.73m^2^. According to medical records, the patient also had chronic lower limb ischemia and stable ischemic heart disease. She had a distal radial-cephalic arteriovenous fistula (RCAVF) created six months before admission. Ultrasound examination revealed that the fistula flow was 170 mL/min, whereas the diameter of the cephalic vein at the elbow crest and in the middle of the forearm was 5 mm and 4.5 mm, respectively. The radial and ulnar arteries were 1.9 mm and 2.4 mm in diameter, respectively. Both vessels exhibited focal atherosclerotic calcifications. In addition, in the distal section of the forearm we discovered a significant narrowing of the RA, accompanied by an acceleration of the wave speed in Doppler, which, as we speculated, was the cause of the fistula maturation failure. Because the cephalic vein was patent, we planned to create a new fistula with the RADAR technique to improve the blood flow through the AVF.

### 2.4. Procedure Description in Case 2

Lignocaine (1%) was used for local anesthesia during the surgery. An oblique incision was made in the middle part of the forearm to dissect the vessels (Figure 4A). The radial artery appeared optimal in this location for creating a fistula, as the Doppler ultrasound revealed a vessel fragment preceding the artery stenosis. The RA was clamped and severed over the ligature (Figure 4A). The vein was then longitudinally cut (Figure 4B), and the artery was transferred to the vein (Figure 4C). The vein and artery were flushed with heparinized saline. Anastomosis was carried out with continuous stitching using a non-absorbable polypropylene vascular suture of 6-0 (Figure 4C). Due to the limited length of the RA, Option II for anastomosis was selected. A weak thrill was felt immediately after the clamps were released. The ultrasound was carried out the following day. The flow rate of the fistula was 234 mL/min. Since the fistula blood flow was not satisfactory, we scheduled the next procedure three weeks later. The patient was readmitted to the clinic once more. In the ultrasound, we identified additional stenosis in the RA near the last anastomosis. The second surgery was performed in a similar fashion, but the incision site was located 2–3 cm proximal to the previous procedure. Figure 4D shows the fistula created with the RADAR technique with significant thrill. An ultrasound examination revealed that fistula function had improved. The following day, the fistula flow rate was 376 mL/min. The vein in the elbow crest increases to 6.9 mm in diameter. The patient was administered salicylic acid in the postoperative period, although this medication was taken prior to the creation of a fistula because of atherosclerosis. Two weeks after the procedure, at the outpatient clinic follow-up visit, the fistula was still patent; however, there were no clinical or laboratory indications to initiate dialysis. The renal function of the patient and the AVF function are still being observed.

### 2.5. Description of Case 3

A 47-year-old woman with chronic kidney disease due to systemic lupus erythematosus and antiphospholipid syndrome on hemodialysis for 10 years was admitted to our clinic for the creation of a new AVF. Nine years ago, she suffered an ischemic stroke. The left brachiocephalic fistula was used for hemodialysis and had required ligation 10 months previously due to skin maceration posing a risk of rupture. At that time, a permanent catheter had been implanted. Due to favorable anatomical conditions, it was decided to create a primary snuffbox fistula on the right side. However, because of the low fistula flow, RCAVF was carried out on the wrist. Unfortunately, a thrombus developed in the middle third of the RA during the postoperative period. In consequence, the fistula flow rate during follow-up visits did not exceed 250 mL/min. Seeing that that the cephalic vein was patent, we decided to create a fistula using the RADAR technique proximal to the RA stenosis.

### 2.6. Procedure Description in Case 3

As in previous cases, the surgical site was determined by the results of the ultrasound examination. In the present case, the patent artery was found in the third proximal of the forearm (Figure 5A). The procedure was carried out as described above. In this case, however, the vein and artery were in close proximity to one another, which is why Option II for anastomosis was chosen. After performing an anastomosis and releasing the clamps, we detected a blood flow in the vessel (Figure 5B). The following day’s ultrasound examination of the fistula revealed a flow rate of 541 mL/min (Figure 6A). The diameter of the radial artery (Figure 6B) and the cephalic vein (Figure 6C) also increased to 4 mm and 7.8 mm, respectively. The patient was given salicylic acid in the postoperative period, despite having taken this medication prior to creating a fistula due to antiphospholipid syndrome. One month later, the fistula was successfully cannulated.

## 3. Results

In each case, the parameters characterizing the fistula’s function were improved (Table 1). Option I for the RADAR technique was selected in Case 1, whereas Option II was used in Cases 2 and 3. No acute complications occurred immediately after all surgeries. In two patients, the arteriovenous fistula (AVF) could be utilized shortly after surgery. In Case 1, the fistula remained patent at the 3-week postoperative follow-up appointment, and the first successful cannulation was performed 4 weeks after surgery (Figure 1). The fistula was still used for hemodialysis two months later. In Case 2, two RADAR interventions were required to achieve a satisfactory AVF function. The residual renal function was sufficient to maintain the patient in predialysis. At the outpatient clinic follow-up appointment two weeks after the procedure, the fistula was patent and an ultrasound examination revealed that it was suitable for cannulation. In Case 3, the fistula was cannulated satisfactorily one month after surgery and was still in use three months later.

## 4. Discussion

In this study, we described a group of patients with delayed fistula maturation due to radial artery stenosis. In three patients, three different causes of radial artery (RA) stenosis were found. RA stenosis was associated with the following causes: anastomosis suturing errors in a primary arteriovenous fistula in Case 1, severe atherosclerosis in Case 2, and RA thrombosis in an active lupus patient as a consequence of antiphospholipid syndrome in Case 3. We adapted a radial artery deviation and reimplantation (RADAR) technique for this specific indication and found that all our patients’ AVF functions improved. To our knowledge, this is the first report of this type of solution.

The failure of AVF maturation is the main barrier to establishing functional arteriovenous fistulas. RADAR is a new concept in AVF creation. Encouraging results reported by Sadaghianloo et al. with high patency rates make this approach interesting [6]. However, RA ligation has been criticized by a number of authors, who point out serious drawbacks, such as an increased risk of hand ischemia and potential complications during fistula salvage in cases of AVF thrombosis, in which emboli propagate into the artery [7]. According to Röhl et al., classical native AVF, i.e., an extended dissection and mobilization of the cephalic vein (CV), which allows the vein to be brought toward the radial artery for an end-to-side anastomosis, may fail as a result of technical errors in suturing the vessels, particularly in the case of small vessel sizes [4]. According to our observations, arterial stenosis during arterial suturing often occurs in the proximal portion of the anastomosis (red point X in Figure 1), which is a narrow segment of the artery and, therefore, susceptible to this type of technical error. At the same time, this stenosis or closure is critical for fistula function, since blood flow from the ulnar artery (UA) via the palmar arches is often inadequate for fistula maturation. In such situations, the maturation period is typically long, and the fistula’s blood flow does not exceed 300 mL/min. In these circumstances, blood flow is reduced in the RA, and, over time, degenerative changes develop as intimal hyperplasia. Due to the alterations in the RA, the repair of such a dysfunctional AVF may provide some difficulties. Our new strategy is to use the RADAR technique as a secondary fistula, which has the following advantages:The primary fistula is not ligated, and the blood flow from the ulnar artery is maintained.By connecting the end of the radial artery to the side of the vein, the blood supply to the cephalic vein is increased, thereby accelerating its maturation. In other words, the additional volume of flow from the RA enhances the total fistula blood flow as measured in the brachial artery.Increased radial blood flow protects against the development of degenerative changes; consequently, a third fistula can be created proximally between the RA and the CV in cases of AVF thrombosis.The risk of hand ischemia is theoretically reduced, as both the UA and the large portion of the RA are patent. The occluded distal part of the RA is bridged by the distal CV (Figure 1). The only limitation is the venous valves, which may impede retrograde flow through the CV. However, this risk is minimal when the venous bridge is not long.We have reported RA stenosis resulting from three possible causes of low fistula flow and maturation failure, but this approach can be used for any type of RA stenosis, including changes after the artery has been used as a vascular access in an intensive care unit or in cardiology procedures. In each of these circumstances, a RADAR procedure may be considered.We believe this technique allows clinicians to avoid upper fistulas involving the brachial artery and graft-based fistulas.

It should be emphasized that one possible disadvantage of this solution is venous hand congestion; however, neither our patients nor the group of RADAR patients in the study by Sadaghianloo et al. noticed this complication [6].

We utilized the RADAR technique in two options (Figure 1). Option I or Option II was selected based on the vascular anatomy. The selection was determined by two factors: first, the length of the dissected RA, and second, the distance between the RA and the CV. The main goal was to create a smooth RA arch with no sharp angles. Option I for anastomosis was chosen due to the long length of the RA and the wide distance between the RA and the CV. This option appears to be advantageous in terms of hemodynamics in fistulas, since direct blood flow from the artery is concordant with vein flow, although this is the operator’s (T.G.) subjective view. This could be a topic for future rheological research. On the other hand, the technique of connecting the vessels in Option I appears subjectively more complicated than in Option II, because sewing the distal angle of the anastomosis was difficult, as the artery covered the walls of the vein in this location, which could lead to technical errors and RA stenosis. In Option II, the anastomosis is similar to that of creating a standard radial-cephalic fistula on the left. That is why the next two patients were operated utilizing Option II. This option has the potential disadvantage that the blood flow is directed in the opposite direction of the blood flow in the vein. Option II was favored by T.G., but individual surgeons may opt for Option I.

## 5. Conclusions

The RADAR fistula may be successfully used as a secondary access in patients with maturation failure caused by RA stenosis to accelerate fistula maturation. This tailored strategy should be evaluated in prospective studies.

## Figures and Tables

**Figure 1 jcm-12-06481-f001:**
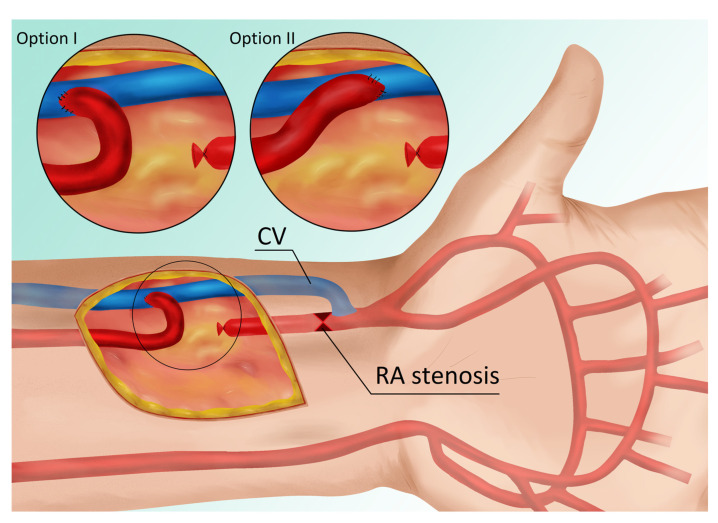
Blood vessels of the hand and vascular connections with two anastomosis options (two magnifications in circles on top). Red point X indicates a radial artery stenosis. Abbreviations: RA—radial artery, CV—cephalic vein.

**Figure 2 jcm-12-06481-f002:**
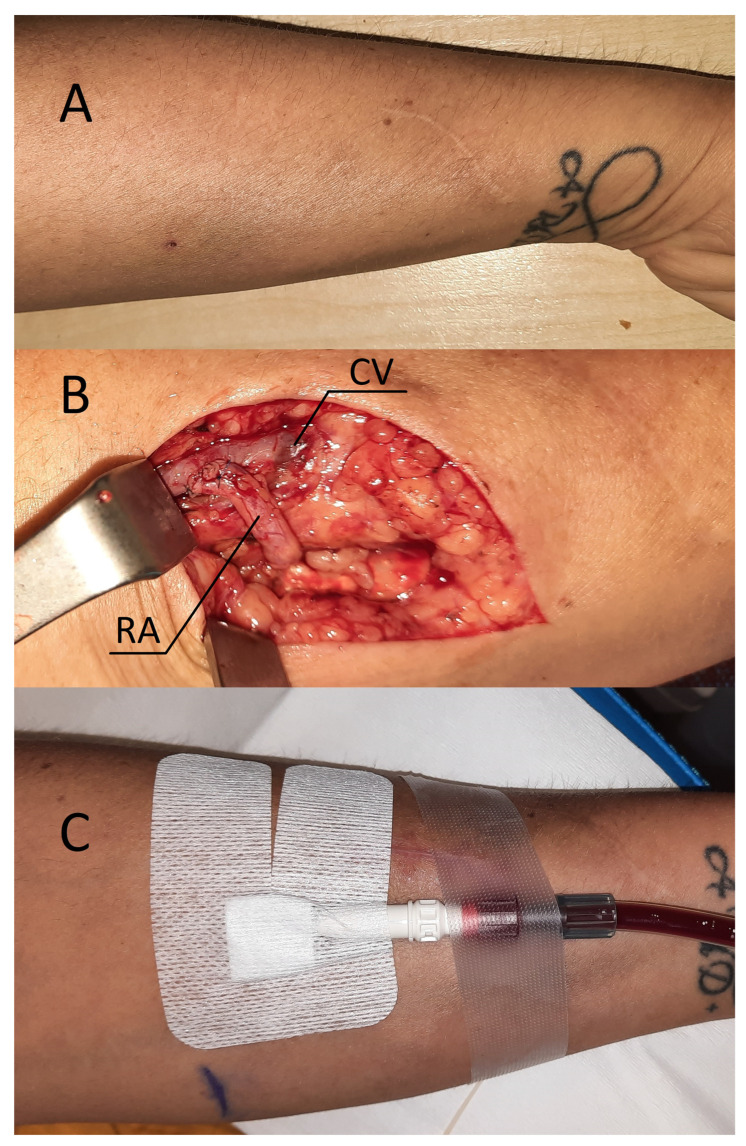
(**A**) Patient’s forearm after classical radial-cephalic fistula. (**B**) RADAR intervention—intraoperative view. (**C**) Successful fistula cannulation and hemodialysis performed by drawing blood from the fistula and returning it via a dialysis catheter. Abbreviations: RA—radial artery, CV—cephalic vein.

**Figure 3 jcm-12-06481-f003:**
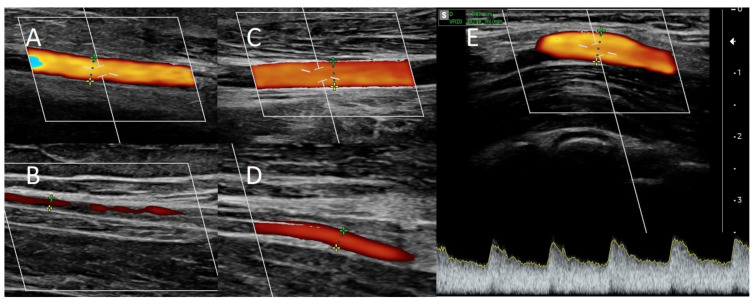
Doppler ultrasound (**A**) of the ulnar artery before RADAR intervention, (**B**) of the radial artery before RADAR, (**C**) of the ulnar artery after RADAR, and (**D**) of the radial artery after RADAR. (**E**) Blood flow in brachial artery after RADAR intervention.

**Figure 4 jcm-12-06481-f004:**
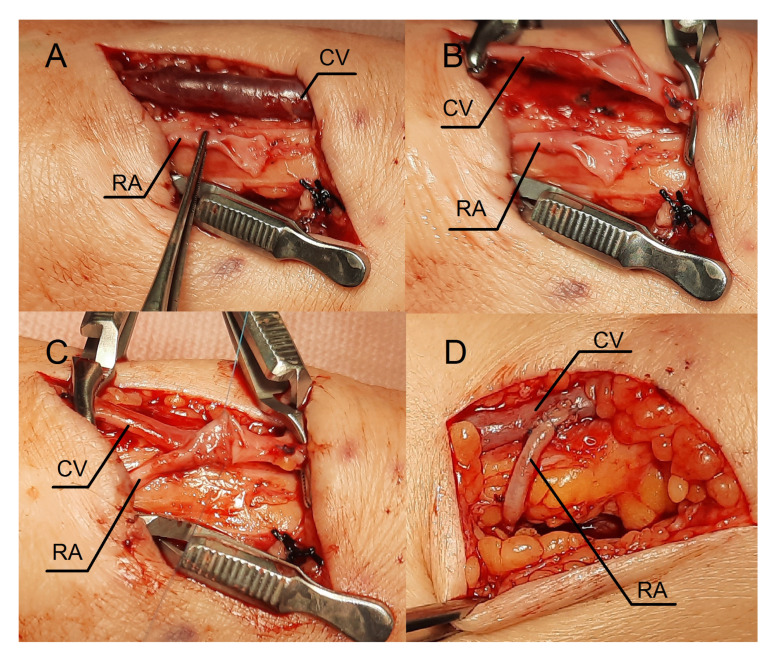
Intraoperative view. Sequential steps in creating the fistula using the RADAR technique. (**A**) An oblique incision in the middle part of the forearm. (**B**) The cephalic vein is incised longitudinally. (**C**) An anastomosis is created using a continuous stitch. (**D**) Intraoperative view during the second RADAR intervention. The thrill was felt immediately after releasing the clamps. Abbreviations: RADAR—radial artery deviation and reimplantation, RA—radial artery, CV—cephalic vein.

**Figure 5 jcm-12-06481-f005:**
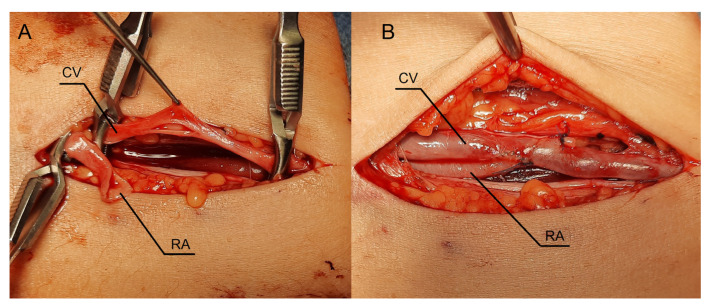
Intraoperative view during creation of the fistula using the RADAR technique. (**A**) A longitudinal incision in the proximal part of the forearm with dissected vessels. (**B**) Stage after releasing the clamps. Abbreviations: RADAR—radial artery deviation and reimplantation, RA—radial artery, CV—cephalic vein.

**Figure 6 jcm-12-06481-f006:**
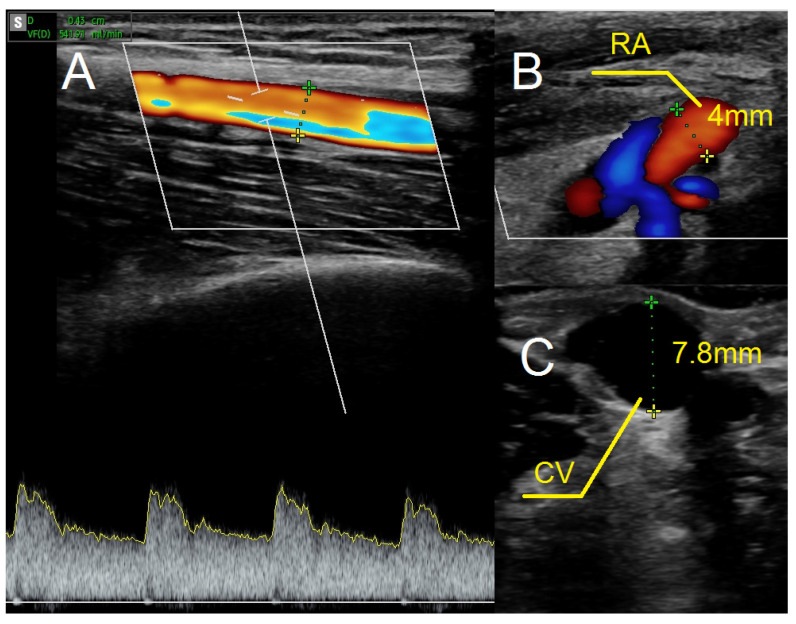
Ultrasound examination after creating the fistula using the RADAR technique. (**A**) Fistula flow measured on brachial artery. (**B**) Doppler ultrasound—a longitudinal section of the radial artery. (**C**) The cephalic vein in cross-section at the cubital fossa. Abbreviations: RADAR—radial artery deviation and reimplantation, RA—radial artery, CV—cephalic vein.

**Table 1 jcm-12-06481-t001:** Ultrasound parameters describing arteriovenous fistula function before surgery and 24 h post-surgery.

	RA Before Surgery (mm)	RA Post-Surgery (mm)	UA Before Surgery (mm)	UA Post-Surgery (mm)	BA Before Surgery (mm)	BA Post-Surgery (mm)	CV * Before Surgery (mm)	CV * Post-Surgery (mm)	Fistula Flow Before Surgery (mL/min)	Fistula Flow Post-Surgery (mL/min)
**Case 1**	1.3	2	4	3.8	5	5.2	5.8	6	264	482
**Case 2**	1.9	2.1 **	2.4	2.4 **	3.6	3.8 **	5	6.9 **	170	376 **
**Case 3**	1.6	4	2.1	2	3.7	4.5	6	7.8	211	541

* The largest diameter of the cephalic vein measured on the forearm. ** The following parameters were measured after the second RADAR intervention. Abbreviations: RA—radial artery, UA—ulnar artery, BA—brachial artery, CV—cephalic vein.

## Data Availability

The data presented in this study are available in this article.

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
