# Peer review of "Radial Artery Deviation and Reimplantation (RADAR) to Accelerate the Maturation of Radial-Cephalic Fistulas for Hemodialysis in Patients with End-Stage Renal Disease"

_jcm, 2023, doi:10.3390/jcm12206481_

Round 1

Reviewer 1 Report

the authors representing a good and common work done for patients with end-stage renal disease undergoing hemodialysis, entitled: (Radial artery deviation and reimplantation (RADAR) to accelerate the maturation of radial-cephalic fistulas for hemodialysis). And I have some comments to polish the article.

TITLE: remove the full stop (.) from the title. Complete the title be as follows: Radial artery deviation and reimplantation (RADAR) to accelerate the maturation of radial-cephalic fistulas for hemodialysis in patients with end-stage renal disease.

MAIN MANUSCRIPT:

Introduction:

Line # (35): not all patients with chronic kidney disease underwent renal hemodialysis, so kindly change this statement to "end-stage renal disease" (ESRD),

Line # (38): a AVF and 15.0% to 20.2% utilized a graft. Please replace the letter (a) to be (an) and correct the remaining part of the sentence to be as follows: an AVF and 15.0% to 20.2% utilized a synthetic arteriovenous dialysis graft. 

Line # (43): (modus Rohl et al.). Please remove the author and co-author’s names from this sentence as the number of the reference is already written (reference # 4).

Line # (58): (In 2016, Sadaghianloo et al.). Please remove the author and co-author’s names from this sentence and write ( it was reported in the literature that favorable outcomes .... ) as the number of the reference is already written (reference # 6) including the author and co-author’s names.

MATERIAL AND METHODS

2.1. Description of case 1.

Line # (72) a catheter-related infection. What due ,meant by catheter; is it a permacath, bilateral femoral catheter etc .... for renal dialysis? Please clarify?

2.2. Procedure description in Case 1. 

Line # (104): insoluble polypropylene. Please replace the word insoluble with non-absorbable. As the word insoluble is applied to fluids only. 

2.3. Description of Case 2.

Lines # (117-118): due to the progression of CKD. Please replace the sentence with "due to the progression to ESRD".

Moderate editing of English language required.

Author Response

Reviewer 1

the authors representing a good and common work done for patients with end-stage renal disease undergoing hemodialysis, entitled: (Radial artery deviation and reimplantation (RADAR) to accelerate the maturation of radial-cephalic fistulas for hemodialysis). And I have some comments to polish the article.

TITLE: remove the full stop (.) from the title. Complete the title be as follows: Radial artery deviation and reimplantation (RADAR) to accelerate the maturation of radial-cephalic fistulas for hemodialysis in patients with end-stage renal disease.

Response: The title has been changed.

MAIN MANUSCRIPT:

Introduction:

Line # (35): not all patients with chronic kidney disease underwent renal hemodialysis, so kindly change this statement to "end-stage renal disease" (ESRD),

Response: The text has been changed.

Line # (38): a AVF and 15.0% to 20.2% utilized a graft. Please replace the letter (a) to be (an) and correct the remaining part of the sentence to be as follows: an AVF and 15.0% to 20.2% utilized a synthetic arteriovenous dialysis graft. 

Response: The text has been changed.

Line # (43): (modus Rohl et al.). Please from this sentence as the number of the reference is already written (reference # 4).

Response: The author and co-author’s names have been removed from the text.

Line # (58): (In 2016, Sadaghianloo et al.). Please remove the author and co-author’s names from this sentence and write (it was reported in the literature that favorable outcomes .... ) as the number of the reference is already written (reference # 6) including the author and co-author’s names.

Response: The author and co-author’s names have been removed and the text was changed.

MATERIAL AND METHODS

2.1. Description of case 1.

Line # (72) a catheter-related infection. What due ,meant by catheter; is it a permacath, bilateral femoral catheter etc .... for renal dialysis? Please clarify?

Response: The patient used permanent dialysis catheter for 16 months. This information has been added to the text.

2.2. Procedure description in Case 1. 

Line # (104): insoluble polypropylene. Please replace the word insoluble with non-absorbable. As the word insoluble is applied to fluids only. 

Response: The text has been changed. 

2.3. Description of Case 2.

Lines # (117-118): due to the progression of CKD. Please replace the sentence with "due to the progression to ESRD".

Response: The text has been modified. We used the phrase “progression of chronic kidney disease to ESRD” because it appears to better reflect the true clinical status of a patient's kidney function.

Comments on the Quality of English Language

Moderate editing of English language required.

Response: The manuscript has been reviewed by a native English speaker.

Thank you for all your linguistic suggestions that helped us improve our manuscript.

Reviewer 2 Report

The manuscript is interesting and has practical value, but needs major correction.

1.     Page 2 line 57. I agree with the author that the closure of the primary anastomosis is disadvantage of this technique, but I do not understand why the distal vein ligation is a disadvantage. 

2.     Page 5 line 106. The authors describe two options of RADAR technique (Fig.1). The text should mention the criteria by which the first and second options were selected. If they are selected intraoperatively and spontaneously  based on anatomy, this should also be noted. 

3.     It would be better if you indicate the follow -up of each patient in results. 

4.     The text does not mention treatment in post-op. Did you use  anticoagulants or antiplatelet drugs? If not, indicate that as well.

5.     Page 9 line 243. Did the authors use maneuver tests to determine the risk of hand ischemia, and if so, which ones?

Author Response

Reviewer 2

The manuscript is interesting and has practical value, but needs major correction.

Response: Thank you for your general opinion regarding the text.

  1. Page 2 line 57. I agree with the author that the closure of the primary anastomosis is disadvantage of this technique, but I do not understand why the distal vein ligation is a disadvantage. 

Response: The Reviewer is right in that distal vein ligation is not clinically significant in many cases of recreation of classical radial-cephalic arteriovenous fistula (due to juxta anastomosis stenosis, for example).  However, in patients with proximal vein stenosis, this distal portion of the vein may be used for cannulation if, for instance, side-to-side anastomosis has been performed and this vein portion with retrograde flow is used for cannulation. This also applied to the RADAR technique, in which the distal vein is not ligated. In these situations, a Doppler ultrasonography examination with flow mapping should be carried out to evaluate the canulation strategy (see our paper: Tomasz Gołębiowski, Mariusz Kusztal, Ewa WÄ…torek, Jerzy Garcarek, Krzysztof Letachowicz, WacÅ‚aw Weyde, Marian Klinger. Consider use of a collateral venous circuit before abandoning the arteriovenous fistula--the experience of a complex vascular access case. Ann Vasc Surg. 2014 Jul;28(5):1320.e9-13. doi: 10.1016/j.avsg.2013.12.014.). It should be noted that one of the disadvantages of side-to-side anastomosis is the difficulty of this technique and the possibility of venous hypertension with extremity oedema.

This discussion has been included in the Introduction section.     

  1. Page 5 line 106. The authors describe two options of RADAR technique (Fig.1). The text should mention the criteria by which the first and second options were selected. If they are selected intraoperatively and spontaneously  based on anatomy, this should also be noted. 

Response:  Thank you for this important inquiry. All fistulas were performed by one operator (TG). Based on vascular anatomy, the anastomotic option was chosen intraoperatively. Two factors played a role in the selection process: first, the length of the dissected radial artery (RA), and second, the distance between the RA and the CV of the cephalic vein. Option I was used in Case 1. It was the case with the longest RA and the widest segment between RA and CV. The primary goal was to achieve a smooth RA arch with no abrupt angles. This option appears to be advantageous in terms of hemodynamics in fistulas since direct blood flow from the artery is concordant with vein flow, although this is the operator’s subjective view. This could be a topic for future rheological research. On the other hand, the technique of connecting the vessels in Option I appears subjectively more complicated than in Option II, because sewing the distal angle of the anastomosis was difficult as the artery covered the walls of the vein in this location, which could lead to technical errors and RA stenosis. In Option II, the anastomosis is similar to that of creating a standard radial-cephalic fistula on the left. That is why two next patients were operated utilizing Option II. This option has the potential disadvantage that the blood flow is directed in the opposite direction of the blood flow in the vein. Option II was favored by TG, but individual surgeons may opt for Option I.

All this information was included in the Material and Methods and Discussion. 

  1. It would be better if you indicate the follow-up of each patient in results. 

Response: We have added more data to the results section.

  1. The text does not mention treatment in post-op. Did you use anticoagulants or antiplatelet drugs? If not, indicate that as well.

Response: During each surgical procedure, the vein and artery were flushed with heparinized saline and approximately 5000 IU of unfractionated heparin was administered per surgery. Two patients (Cases 2 and 3) received permanent salicylic acid also in the postoperative period. In Case 2, salicylic acid was administered due to atherosclerosis, and in Case 3 due to antiphospholipid syndrome.

This information was included in the Material and Methods (Description of the Cases).

  1. Page 9 line 243. Did the authors use maneuver tests to determine the risk of hand ischemia, and if so, which ones?

Response: Yes, All patients underwent a preoperative clinical examination and Doppler ultrasound venous and arterial mapping. Prior to any procedure that could affect the patency of the radial artery, we used a Doppler ultrasound and a modified Allen test to evaluate the patency of the ulnar artery and palmar arches. In all patient we additionally carried out the Doppler Allen's Test and found the flow in the superficial palmar branch, dorsal digital thumb artery and ulnar artery with radial artery compression indicating low risk of hand ischemia.

We added this information in the Material and Methods section.

Thank you for all remarks that helped us improve our manuscript.

Round 2

Reviewer 2 Report

I have no more comment and questions. The authors convincingly answered my question and made correction to the manuscript.